



# Operationalizing equity in multipurpose water systems

Guang Yang[1], Matteo Giuliani[1], Andrea Castelletti[1]

[1]Department of Electronics, Information, and Bioengineering, Politecnico di Milano, Milan, 20133, Italy.

*Correspondence to*: Matteo Giuliani (matteo.giuliani@polimi.it)

**Abstract** Participatory decision-making is a well-established approach to address the increasing pressure on water systems induced by growing multi-sectoral demands and increased competition among different water users. Yet, most existing approaches search for system-wise efficient solutions and do not quantify their distributional effect among the stakeholders. In this work, we investigate how to operationalize equity principles to design improved water systems operations that better balance efficiency and justice. More specifically, we explore to which extent the inclusion of equity principles reshapes the

space of efficient solutions. Numerical experiments are conducted on the Lake Como system, Italy, operated primarily for flood control and irrigation water supply while also providing recreation and river ecosystem services. Our results show how incorporating equity considerations into the design of water system operations enriches the solution space by generating more compromise solutions than those obtained using a traditional multi-objective optimization. Moreover, we find that including equity in the operating policy design can indirectly improve the performance of marginalized sectors, such as recreation and

ecosystem, which are not explicitly considered by the current lake operation. Lastly, we illustrate how the aggregation of multi-sectoral interests into an equity index strongly shapes our results. Eliciting the preference structure of stakeholders and policymakers thus becomes paramount for the identification of a fair balance across competing interests. This work bridges the gap between multi-objective optimization approaches and equity-informed decision-making for real-world water resources planning and management, providing an effective tool to promote efficient and equitable policies.

## 1. Introduction

Proper operation of existing water systems is widely recognized as one of the most important and cost-effective ways to improve water use efficiency and reduce stresses caused by rapid population growth and socio-economic development (Benson, 2016;Gaupp et al., 2015;Ho et al., 2017;Rodell et al., 2018;Wallington and Cai, 2020). Water reservoirs generally serve multiple and competing purposes, including flood control, irrigation, power generation, navigation, and river ecosystem

maintenance, but limited water resources make it impossible to fully and simultaneously satisfy all these water users (Billington and Jackson, 2006;Groenfeldt, 2019). At the same time, growing energy and food demands are putting additional pressure on these systems and are exacerbating conflicts (Ehsani et al., 2017;Giuliani et al., 2016a;Olsson, 2015), which are often related to emerging adverse social and environmental consequences caused by water infrastructure (Graham et al., 2020;Poff et al., 2016;Poff and Schmidt, 2016;Sabo et al., 2017;Schmitt et al., 2018) and to water security (D'Odorico et al., 2018;Liu et al.,

2018;Scanlon et al., 2017). These challenges represent a well-established topic in the water systems analysis community since the Harvard Water Program (Maass et al., 2013), promoting the idea of adopting an a-posteriori decision making based on trade-off analysis between competing objectives (Cohon and Marks, 1975;Maier et al., 2014;Nicklow et al., 2010;Reed et al., 2013).

Traditionally, water system operations are formulated as multi-objective decision-making problems and the underlying

conflicts among objectives capturing the interests of different stakeholders yield a set of Pareto optimal (or efficient) solutions rather than a single optimal solution (Loucks and Van Beek, 2017). A solution is defined as being Pareto optimal (or nondominated) if no other solution gives a better value for one objective without degrading the performance in at least one other objective. In this context, most existing approaches search for the Pareto optimal set to explore trade-offs between





operating objectives (Geressu and Harou, 2019;Giuliani et al., 2014;Kasprzyk et al., 2009;Schmitt et al., 2018). Yet, Pareto
optimality pursues system-wise efficiency and ignores the distributional effects of the optimal solutions among the different
stakeholders, potentially resulting in inequitable outcomes. This potential inconsistency between efficiency and equity might
inadvertently bias the analysis on efficient but unfair solutions that the stakeholders will hardly accept. Including equity among
the objectives can be useful to ensure that the negotiations on the solution to be implemented succeed smoothly.

There is a growing interest in equity-related research in the water resources planning and management literature. For example,
Wang et al. (2008) developed a cooperative water allocation model to achieve fair and efficient water allocation among
competing stakeholders at the basin level. Girard et al. (2016) designed cost-effective and equitable portfolios for water
resource adaptation to climate change in the Orb River basin by implementing cooperative game theory and social justice
approaches. Siddiqi et al. (2018) developed a set of reliability and equity metrics to quantitatively evaluate the water security
in a canal irrigation system in the Indus basin. Ciullo et al. (2020) proposed a decision criterion to account for the geographical
distribution of flood risk in the transboundary area of the German-Dutch Lower Rhine River and investigate the impact of
equity criteria on flood risk management. Most previous equity-related studies in water resources mainly focus on evaluating
equity and how to promote equity by fairly distributing benefits or risks. Some recent literature has proposed that equity can
be improved by integrating it into water resources operation design. For example, Gullotta et al. (2021) improved the equity
among users of a water distribution network in northern Italy by optimizing the placement of control valves to maximize the
uniformity coefficient (an equity index proposed in (Gottipati and Nanduri, 2014)). Kazemi et al. (2020) optimized the water
allocation in the Sefidrud basin in Iran for the maximum water use revenue and minimum Gini index (which was introduced
by Gini (1921) to measure income inequality). Although these works promote equity in water resource operation, the impacts
of equity on multi-objective decision-making and how equity affects the trade-offs between conflicting objectives have not yet
been studied.

In this work, we investigate how to operationalize equity principles into the multi-objective design of improved water systems
operations that better balance trade-offs among competing stakeholders' interests. The approach is demonstrated on the Lake
Como system, a regulated lake in Northern Italy historically operated for flood protection and irrigation supply. Over the last
few years, the increasing frequency and intensity of severe droughts emphasized the importance of additional, so far
marginalized services provided by the lake operations, such as preventing low levels from supporting recreational activities
and ecosystem preservation downstream of the lake.

The paper provides two main contributions. First, we analyze alternative problem formulations to assess how the inclusion of
equity principles reshapes the space of efficient solutions with respect to both a traditional optimization considering only
primary objectives and an inclusive optimization that account for primary as well as historically marginalized objectives.
Second, we explore the sensitivity of the resulting solutions with respect to the definition of the equity metric, which, in a
multi-objective problem, requires the aggregation of multiple objective functions into a single index (e.g., coefficient of
variation). However, aggregated objective functions might adversely bias the designed alternatives in unpredictable ways
(Kasprzyk et al., 2016). This is an important aspect to explore the operationalization of equity in multipurpose water systems.

The rest of the paper is organized as follows: in the next section, we introduce the Lake Como study site, while Sect. 3 describes
the adopted methodology. Then, results and discussion are reported in Sect. 4, while conclusions and final remarks are
presented in the last section.





## 2. Study Site

Lake Como, also known as Lario, is the third-largest lake in Italy and the fifth deepest lake in Europe; it has an active storage capacity of 254 million $m^3$ and a depth of over 400 m. The catchment area of Lake Como is approximately 4,552 $km^2$, with the lake serving an irrigation-fed cultivated area of about 1,300 $km^2$ (Fig. 1). The major crops within the agricultural area
include cereals, maize, and temporary grasslands for livestock. The climate of Lake Como is temperate around the lake and cold in the upper alpine catchment (Peel et al., 2007). The hydrologic regime is snow-rainfall dominated, with the dry seasons in winter and summer and wet seasons in late spring and autumn, respectively.

The lake's shape is close to an inverted letter "Y" and the city Como is located at its southwestern branch. Because of the dead-end and the lowest elevation on the lake shoreline at Como, this area is prone to flooding. Thus, the regulation of Lake Como
has been historically studied mostly by looking at the conflict between irrigation water supply and flood control (Denaro et al., 2017;Giuliani et al., 2020;Guariso et al., 1986). Spring/summer snowmelt primarily creates the seasonal storage of Lake Como, which can be reallocated to satisfy the summer water demand peak for irrigation. Storing more water in spring will benefit the irrigation water supply in summer; however, this strategy could lead to high lake levels for longer periods and thus increase flood risk.

Lake Como is also a popular tourist destination because of its beautiful Alpine landscape and abundant wildlife, and it is a scenic spot for sailing, boating, and windsurfing. Interests related to ecosystems, tourism, navigation, and fishing are also attracting more and more attention in Lake Como water governance in recent years (Carvalho et al., 2019;Grizzetti et al., 2016). Accordingly, the Lake Como operation design problem can be formulated as a problem that involves up to four competing objectives, where recreation services (e.g., tourism and navigation) and river ecosystem maintenance downstream of the lake
are added to irrigation water supply and flood control.

## 3. Methods and Tools

### 3.1. Model description

The model of the system reproduces the dynamics of Lake Como by using a mass-balance equation of the lake storage $s_t$ ($m^3$) assuming a modeling and decision-making time step $\Delta t = 24$ hours, where the lake releases are determined by the lake operating
policy:

$$s_{t+1} = s_t + (q_{t+1} - r_{t+1}) \cdot \Delta t \qquad (1)$$

where $q_{t+1}$ ($m^3$/s) and $r_{t+1}$ ($m^3$/s) are the net inflow to the lake and the actual lake release in the time period [$t$, $t$+1), respectively. Specifically, the release volume $r_{t+1}$ is determined by a non-linear, stochastic function that depends on the release decision $u_t$ (Soncini-Sessa et al., 2007) and accounts for the effect of the uncertain inflows between the time $t$ (at which the decision is
taken) and the time $t$+1 (at which the release is completed). The release $r_{t+1}$ could not be equal to the decision $u_t$ due to existing legal and physical constraints on the lake level and release (e.g., spills, dead storage). According to the daily time step, the Adda River can be described by a plug-flow model to simulate the transfer of the lake releases from the lake outlet to the intake of the irrigation canals. The water diversions from the Adda River into the irrigation canals are regulated by the water rights of the agricultural districts.

The lake operating policies that determine the release decision $u_t$ are defined as Gaussian radial basis functions (RBFs; (Bușoniu et al., 2011)) as follows


$$u_t = \alpha + \sum_{k=1}^{K} \omega_k \varphi_k(X_t) \quad t = 1, \dots, H \quad 0 \le \varphi_k \le 1 \tag{2}$$

$$\varphi_k(X_t) = exp\left[-\sum_{j=1}^{M} \frac{((x_t)_j - c_{j,k})^2}{b_{j,k}^2}\right] \quad c_{j,k} \in [-1,1], b_{j,k} \in (0,1) \tag{3}$$

where $K$ is the number of RBFs, $\omega_k$ is the weight of the $k^{th}$ RBF, $M$ is the number of input variables $X_t$, and $\mathbf{c}_k$ and $\mathbf{b}_k$ are the

$M$-dimensional center and radius vectors of the $k^{th}$ RBF, respectively. Lake level $h_t$, previous day inflow $q_t$, and the day of the year $\tau_t$ are used as input variables (i.e., $X_t = (s_t, q_t, \tau_t)$), and the number of RBFs is set to four ($K=4$). The final parameters vector can be summarized as: $\theta = [\alpha, \omega_k, c_{j,k}, b_{j,k}]$, and it thus contains 29 parameters (decision variables) to determine the release decision $u_t$. The operating policies are then optimized using the Evolutionary Multi-Objective Direct Policy Search (EMODPS) method (Giuliani et al., 2016b), a Reinforcement Learning approach that combines direct policy search, non-linear

approximating networks, and multi-objective evolutionary algorithms.

**3.2. Operating objectives**

Building on previous works (Galelli and Soncini-Sessa, 2010;Giuliani and Castelletti, 2016;Giuliani et al., 2016c;Zaniolo et al., 2021), we formulate four objectives capturing the competing interests introduced in the previous section as follows

    (a) Flood control: the high-level reliability (to be maximized) defined as


$$J^F = 1 - \frac{n^F}{H} \tag{4}$$

where $n^F$ is the number of days in the evaluation horizon $H$ during which the lake level is above a flood level threshold.

    (b) Irrigation water supply: the daily average volumetric reliability (to be maximized) defined as

$$J^I = \frac{1}{H}\sum_{t=1}^{H}\left(min\left(\frac{Y_{t+1}}{w_t}, 1\right)\right) \tag{5}$$

where $Y_{t+1}$ (m³) is the daily volume of water available for irrigation, subject to the minimum environmental flow constraint to

ensure adequate environmental conditions in the Adda River downstream of the abstraction point, and $w_t$ (m³) is the corresponding irrigation demand.

    (c) Recreation services: the low-level reliability (to be maximized) defined as

$$J^R = 1 - \frac{n^R}{H} \tag{6}$$

where $n^R$ is the number of days in the evaluation horizon $H$ during which the lake level is below a time-varying, low-level

threshold equal to the $10^{th}$ percentile of the historical lake level.

    (d) River ecosystem: the reliability of environmental flow (to be maximized) defined as

$$J^E = \frac{1}{H}\sum_{t=1}^{H} g(r_{t+1}) \tag{7}$$



where the function $g(r_{t+1})$ returns 1 if $q_t^n - \sigma_t^n \leq r_{t+1} \leq q_t^n + \sigma_t^n$, with $q_t^n$ and $\sigma_t^n$ representing the mean and standard deviation, respectively, of the Adda river flow in natural conditions, otherwise it returns 0.

### 3.3. Operationalizing equity

The equity index considered in this study is formulated as in Siddiqi et al. (2018) as the ratio between the standard derivation ($\sigma$) and mean ($\mu$) of the performance in the four objectives introduced in the previous section, i.e.

$$\zeta = \frac{\sigma(J^F, J^I, J^R, J^E)}{\mu(J^F, J^I, J^R, J^E)} \quad (8)$$

The lower the value of $\zeta$, the more equitable the solution is, with low values of $\zeta$ obtained for high values of the original objectives with a limited performance dispersion across the four objective functions. When the objectives capturing diverse stakeholders' interests are expressed in different units of measure or explore different performance ranges, it could be necessary to map the original objectives into a satisfaction value by applying a value function. The latter allows re-scaling all the objectives into a dimensionless scale, e.g., from 0 to 1, by means of a linear or non-linear transformation. Assessing the equity index by computing the mean and standard deviation of satisfaction values rather than objectives is expected to improve the analysis by working on commensurable quantities.

### 3.4. Experimental settings

As mentioned before, the Lake Como operator traditionally considers two primary objectives (irrigation water supply and flood control). More recently, other needs such as recreation services and river ecosystem maintenance are emerging due to increasingly frequent droughts, which motivates us to investigate how to fairly account for these previously marginalized objectives into the policy design. In this work, we contrast four rival formulations of the Lake Como EMODPS problem (Table 1) that can be summarized as follows:

- P1 – traditional formulation: $\theta^* = \arg\max_\theta \mathbf{J}(\theta) = |J^F, J^I|$

- P2 – traditional & fair formulation: $\theta^* = \arg\max_\theta \mathbf{J}(\theta) = |J^F, J^I, -\zeta|$

- P3 – inclusive formulation: $\theta^* = \arg\max_\theta \mathbf{J}(\theta) = |J^F, J^I, J^R, J^E|$

- P4 – inclusive & fair formulation: $\theta^* = \arg\max_\theta \mathbf{J}(\theta) = |J^F, J^I, J^R, J^E, -\zeta|$

P1 is a traditional multi-objective optimization problem that only searches for the maximum of the two primary objectives. P3 is an inclusive optimization that instead considers all four objectives. Finally, P2 and P4 add the equity index from equation (8) to the traditional and inclusive formulations, respectively. While the comparison between the traditional and inclusive formulations will provide the benefit of including all objectives into the policy design, the comparison between P1 vs. P2 and P3 vs. P4 will focus on assessing the value of including an equity-related objective function in either the traditional multi-objective optimization and the inclusive optimization problems. Finally, the comparison between P2 and P3 allows investigating the differences between using the equity index as a means for indirectly including in the problem formulation the traditionally marginalized objectives or formulating an inclusive optimization that directly includes all stakeholders' interests as separated objectives.





**Table 1 Summary of the alternative problem formulations.**

| Problem | Formulation | Objectives | Including equity (YES/NO) |
|---------|-------------|------------|---------------------------|
| **P1** | Traditional | $J^F, J^I$ | NO |
| **P2** | Traditional & fair | $J^F, J^I, \zeta$ | YES |
| **P3** | Inclusive | $J^F, J^I, J^R, J^E$ | NO |
| **P4** | Inclusive & fair | $J^F, J^I, J^R, J^E, \zeta$ | YES |

The parameters in RBFs-based policies are optimized using the Borg MOEA evolutionary algorithm (Hadka and Reed, 2013), which proves highly robust in solving many-objective control policy optimization problems (Salazar et al., 2016). The number of the function evaluations is 2 million, the same as in previous Lake Como operation optimization (Denaro et al., 2017). To ensure solution diversity and reduce the impact of stochastic factors on the optimal solutions, each optimization was randomly repeated ten times (i.e., the final set of nondominated solutions for each problem are obtained from 10 random optimization trials). In total, the analysis comprises 80 million simulations that required approximately 600 computing hours on an Intel Xeon E5-2660 2.20 GHz with 32 processing cores and 96 GB RAM.

## 4. Results and Discussion

### 4.1. Multi-objective optimization and equity operationalization

The optimization results of problems P1-P4 can be evaluated in terms of four operation objectives along with the equity index using the parallel coordinates plots in Fig. 2, where each line crossing multiple axes represents one Pareto optimal solution. The leftmost axis represents different problem formulations (e.g., the value "2" refers to P2; apart from the axis, the line color is also used to differentiate various problem formulations), and other axes represent solution performance in terms of flood control, irrigation water supply, recreation, environment, and equity. The axis for equity is reversed to ensure that the direction of preference is always upward, and the ideal solution would thus be a horizontal line at the top of each plot. The diagonal lines between adjacent axes infer the conflicts between different objectives.

According to Fig. 2 (a), different problem formulations generate diverse solution spaces. P1 attains good performance in the objectives $J^F$ and $J^I$ (up to 0.99 and 0.91, respectively) that this formulation is optimizing, but low performance on the non-optimized objectives $J^R$ and $J^E$ (lower than 0.67 and 0.72, respectively). Consequently, the equity of these solutions across the four objectives is low. Moving from P1 to P2 allows the attainment of better equity values, which, however, induces performance degradation in terms of $J^F$. The inclusive optimization supports the full exploration of the trade-offs between the four objectives, remarkably amplifying the trade-offs between $J^R$ and $J^F/J^I$ (i.e., the maximum performance in $J^R$ is equal to 1, while the worst solution in flood and irrigation supply is much lower than with P1 or P2). Notably, the solutions of P3 attain lower values of equity than the solutions of P2. Lastly, moving from P3 to P4 produces a small improvement in terms of equity with minor differences in the performance across the four objectives.

Figure 2 motivates investigating how the considered equity metric changes across the different sets of solutions. Fig. 3 shows that the values of the equity index vary mainly with the standard deviation $\sigma$ of the performance in terms of the four objectives instead of the corresponding mean $\mu$: the equity index consistently increases with $\sigma$, but can have different values for the same $\mu$ value, especially in the situation of inequity. The reason is that the trade-off between different objectives can lead to notably different $\sigma$ but similar $\mu$ for two different solutions (e.g., in Fig. 2 (b), high values of $J^I$ generally correspond to low values of $J^R$). The results in Fig. 3 show that the maximum $\mu$ value increases from 0.81 for P1 to 0.90 and 0.91 for P2 and P3, respectively. The $\mu$ here can be considered a proxy of overall performance, and its significant increase indicates the advantages of solutions from P2 and P3 over P1. Yet, higher $\mu$ does not precisely refer to a better solution as the profits per unit increment of the four



objectives are different. In the traditional formulation, the non-optimized objectives introduce variability in system-wise performance that leads to low equity. Conversely, P2 optimizes the equity index computed across the 4 objectives, and this generates a substantial improvement at the system level because of the indirect consideration of the marginalized objectives in the policy design. When transiting from P3 to P4, the equity index instead conveys smaller additional information, so the advantage of P4 over P3 is less evident.

The density distribution of the solutions from different problems (Fig. 4) can be used to investigate further the consequences of adopting alternative problem formulations. Apparently, the solution space generally increases with the number of optimization objectives: for example, the values of $J^F$ and $J^I$ in the traditional formulation (with 2 objectives) are more concentrated (with most of the solutions ranging from 0.95 to 1) than in P2 (with 3 objectives), with the latter that are more concentrated than in P3 (with 4 objectives) (see Fig. 4 (a) and (b)). It is worth noting that the solutions' distribution for P3 and P4 (with 4 or 5 objectives) are very similar because the additional equity objective in the inclusive & fair formulation is highly correlated with all other objectives.

The results in Fig. 4 show that P2 outperforms P1 in terms of all objectives except $J^F$, with large improvements in the traditionally marginalized objectives of $J^R$ and $J^E$ while having similar performance in $J^I$. This asymmetric result can be explained because a good performance in $J^F$ is attained by keeping low lake levels in order to minimize the flood risk (see also Fig. 5), with this condition that negatively impacts the other three objectives. To minimize the standard deviation across the four objectives while maximizing their average values, P2 hence tends to generate solutions that keep relatively high-water levels to increase the performance in $J^I$, $J^R$, and $J^E$, and consequently increase the value of $\mu$, while accepting a reduction in terms of $J^F$ that is anyway positively contributing in terms of equity by reducing $\sigma$.

Interestingly, P2 generates solutions that perform very well in terms of $J^E$ although this objective is not directly included in the optimization, but only indirectly considered through the minimization of $\zeta$. This happens because the upper bound of $J^E$ in P1 is much lower than other objectives, and thus the equity index tends to decrease as the value of $J^E$ increases. Optimizing equity can therefore be a way for improving the objective(s) of marginalized stakeholders with the lowest level of satisfaction (e.g., the $J^E$ with the lowest upper bound in this case study).

The impacts of problem formulations on the resulting dynamics of Lake Como can be evaluated from the density distribution comparisons of median, 25th percentile, and 75th percentile (representing normal, dry, and wet conditions, respectively) of lake water levels (see Fig. 5). P1 tends to get lower water levels than other problem formulations because it only focuses on flood control and irrigation water supply. The reduction of flood risk suggests maintaining low water levels during the wet season (to maximize $J^F$), and satisfying irrigation water demand (to maximize $J^I$) further decreases water levels during the dry summer. Both P2 and P3 tend instead to produce higher water levels than P1 (see Fig. 5 (a)) because high water levels could deteriorate $J^F$ but will improve the performance in other objectives. For example, a high-water level at the end of the flood season provides enough water for the recreation and environmental water release requirements in the coming dry season.

Although both P2 and P3 get higher lake levels than P1, the highest water level of the inclusive formulation is always above that of the traditional & fair one. More specifically, according to Fig. 5 (b), solutions from P3 have the highest 75th percentile of water level close to the flood level threshold (1.24 m), while the highest 75th percentile of the water level from P2 is approximately 1.2 m. By searching for the full range of trade-offs between the four objectives without considering the equity index, the inclusive optimization emphasizes more $J^F$-$J^R$ conflict and design solutions that perform exceptionally well in one objective (e.g., recreation water level maintenance $J^R$) but very poorly in the other one (e.g., flood control $J^F$). These solutions, however, will also attain low levels of equity due to a high standard deviation between the objectives' values. In contrast, P2





avoids exploring such extreme solutions by keeping lower lake levels in wet conditions (see Fig. 5 (b)) and higher water levels in dry conditions (see Fig. 5 (c)). This confirms the potential discrepancy between efficiency and equity. The inclusive optimization better supports the full exploration of trade-offs but generates less equitable solutions, while the traditional & fair formulation yields to more equitable operating policies without exploring the full range of trade-offs. The inclusive & fair formulation emerges as a win-win approach able to simultaneously explore the full range of trade-offs and lake dynamics as P3 (see Fig. 4 and Fig. 5) as well as to attain the best performance in terms of equity (see Fig. 2).

**4.2. Impacts of objectives' aggregation into the equity index**

The ranges of performance attained by the different sets of solutions across the four objectives are not identical (see Fig. 2 (a)). More specifically, the obtained minimum and maximum values of $J^F$, $J^I$, $J^R$, and $J^E$ across the different problem formulations are 0.69 and 0.99, 0.84 and 0.91, 0.53 and 1.00, and 0.63 and 0.88, respectively. According to the equity index considered in this study (see equation (8)), the most equitable solution is obtained when all objectives have the same performance. Assuming,

for example, that a fair solution was a policy having a 0.875 reliability in all objectives, this would imply that $J^E$ would be highly satisfied as the 0.875 is close to the maximum value of 0.88; conversely, the performance in $J^F$ would be only intermediate as the 0.875 is still much lower than the maximum value of 0.99. Thus, using the equity index directly computed by aggregating the reliability performance across the four objectives may lead to some bias in the equity assessment, especially when the minimum and maximum performance values are significantly different.

Applying a value function that transforms each objective into a satisfaction value defined over a homogenous 0-1 scale can, to some extent, correct this bias. To explore the sensitivity of our results with respect to the use of different value functions in the formulation of the equity index, in this work, we examine the impacts of adopting both a linear and a non-linear value function (see Fig. 6). The linear value function (Fig. 6 (a)) maps a performance value $x$ to $x'$ in the range [0, 1] where these extremes correspond to the lowest and highest values of reliability across all the solutions, respectively. In the non-linear value

function (Fig. 7b), we hypothesized a function that requires high values of $J^F$ and $J^I$ to get high values of satisfaction in these objectives as the historical operation of the Lake Como is primarily looking at flood control and irrigation water supply while accepting lower levels of reliabilities in terms of $J^R$ and $J^E$. It should be noted that the purpose of this experiment is only testing the sensitivity of the equity index to the use of different values functions rather than capturing the preference structure of real stakeholders.

The results of this analysis are reported in Fig. 8(a) for the traditional & fair formulation, where each line represents the performance of the optimized solution in terms of $J^F$, $J^I$, $J^R$, $J^E$ (as in Fig. 2) along with three equity indexes computed with either the original values of reliability or the satisfaction values returned by the linear and non-linear value functions. Since the performance difference between linear and non-linear value functions is less notable for P4, the corresponding comparisons are not reported. According to Fig. 7 (a) (violet lines), using standard equity yields solutions with a high level of $J^E$, which can

be explained by the low upper bound of $J^E$ (as aforementioned in the analysis of Fig. 4 in Sect. 4.1). More specifically, because the maximum value of $J^E$ is much lower than other objectives, further increment of $J^E$ based on its high levels of $J^F$ and $J^I$ will decrease the values of the standard equity index. Similarly, using the equity index computed on satisfaction values tends to obtain a high level of performance in terms of $J^F$ due to the upper bound of $J^F$, leading to relatively lower $J^E$ but higher $J^F$ than using the standard equity. Furthermore, using a linear or non-linear value function also affects the equitable operating policy

design. The non-linear value function tends to get relatively higher $J^F$ and $J^R$ but lower $J^E$ than the linear value function because steeper slopes in value function curves occur in a high level of $J^F$ and $J^R$ (i.e., the improvement of $J^F$ and $J^R$ will be considered more important than the improvement of $J^E$). Thus, the preferences of stakeholders and decision-makers should be embedded



in the equity index formulation. More specifically, the optimization of an equity index is able to consider the stakeholders' preferences through a value function by getting better performance in terms of more important objectives.

Figure 7 illustrates the 5% most equitable solutions with respect to the different formulations of the equity index to better analyze the results. The nearly horizontal lines between axes $J^F$ and $J^E$ in Fig. 7 (b), (c), and (d) represent solutions attaining an equity index close to zero. Solutions using standard equity tend to get higher $J^E$ (close to 0.88) but lower $J^F$, $J^I$, and $J^R$ than using normalized equity. When $J^F$, $J^I$, and $J^R$ are higher than 0.88, decreasing the standard equity index will always increase the $J^E$ but may deteriorate other objectives (i.e., there will exist a clear trade-off between equity index minimization and $J^F$, $J^I$,

and $J^R$ maximization). Standard equity leads to all objective performance values close to 0.88 (Fig. 7 (b)). In contrast, linearly and non-linearly normalized equity yields more evenly distributed performance scores. The horizontal red lines in Fig. 7 (c) show all objectives having the same linearly normalized score as the y-axis limits of Fig. 7 (c) are actually the objective ranges, while cyan horizontal lines in Fig. 7 (d) mean the same non-linearly normalized score for all objectives. Also, using linearly normalized equity to some degree improves $J^I$ (Fig. 7 (c)), while using non-linearly normalized equity tends to get more

solutions with a high level of $J^F$ than linearly normalized equity, which is especially notable from the comparisons of performance scores in Fig. 7 (d). It is worth noting that the mean of all objective values from solutions using these three types of equity index seems close to each other (non-linear normalization gets higher $J^F$ and $J^R$ but lower $J^I$ and $J^E$ than linear normalization, no normalization tends to get relatively higher $J^E$ but lower $J^F$, $J^I$, and $J^R$), which further indicates that a low equity index is achieved mainly by lowering the standard deviation of the performance instead of increasing the mean of

objective values (this finding is in accordance with the results in Fig. 3).

## 5. Conclusions

In this paper, we incorporated equity principles into the operation design of multipurpose water reservoirs. Using the real-world case study of Lake Como in Northern Italy, the potential for operationalizing equity indexes is assessed by means of a rival framings experiment where we compare the solutions obtained formulating alternative problems with a different number

of objectives. Moreover, we assess the sensitivity of the proposed approach with respect to the value functions adopted for aggregating the different objectives in the computation of the equity index.

The comparison between operating policies designed with and without considering equity amongst the operating objectives shows that (1) including equity in the operation design can indirectly improve marginalized objectives that are not explicitly considered in the optimization problem; (2) when also explicitly including marginalized objectives, still the addition of an

equity indicator generates more solutions mitigating the conflicts between the operating objectives.

Our work also emphasizes that the search for equitable solutions across multisectoral interests depends on how the multiple objectives are combined to formulate the equity index. Results show that using an equity index based on the original reliabilities can favor or negatively impact some objectives in a difficult way to control. The adoption of participatory approaches for eliciting the preference structure of stakeholders and policymakers thus becomes paramount for the operationalization of equity

principles to re-scale the objectives and represent a fair balance across competing interests.

The definition of equity is not unique in the literature. Beside the equity index used in this study, it could be interesting to investigate the impacts of alternative definitions of equity on water resources decision-making and how to select an appropriate equity metric for a specific problem. Moreover, although equitable solutions can help to mitigate the conflicts among multiple objectives, it is still not easy to design an objective value function and thus choose an equitable policy agreed by all

stakeholders thus further research is required to formulate guidelines for the identification of satisfactory alternative from the





set of Pareto optimal solutions. Finally, the equity in this study is assumed static throughout all experiments, but it could dynamically change over time according to the potential evolution of stakeholders' and decision makers' preferences. It would be interesting to evaluate the dynamics of equity under varying conditions, including future climate scenarios and modifications in the irrigation systems.

**Code and data availability.** Observations of lake inflows were provided by Consorzio dell'Adda (http://www.addaconsorzio.it, Consorzio dell'Adda, 2020). The source code for the Lake Como simulation and EMODPS implementation is available on GitHub (https://github.com/EILab-Polimi/LakeComo).

**Author contributions**. GY, MG, and AC designed the research. GY conducted the numerical experiments and led the data analysis, including the production of the figures in the paper. MG and AC contributed to the analysis of results. All authors
were involved in the writing of the paper.

**Competing interests**. The authors declare that they have no conflict of interest.

**Financial support**. The work has been supported by the IN-WOP - Mind the Water Cycle Gap: Innovating Water Management Optimisation Practice research project, funded by the Water Joint Programming Initiative 2018 (grant n. JPI18_00104).

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



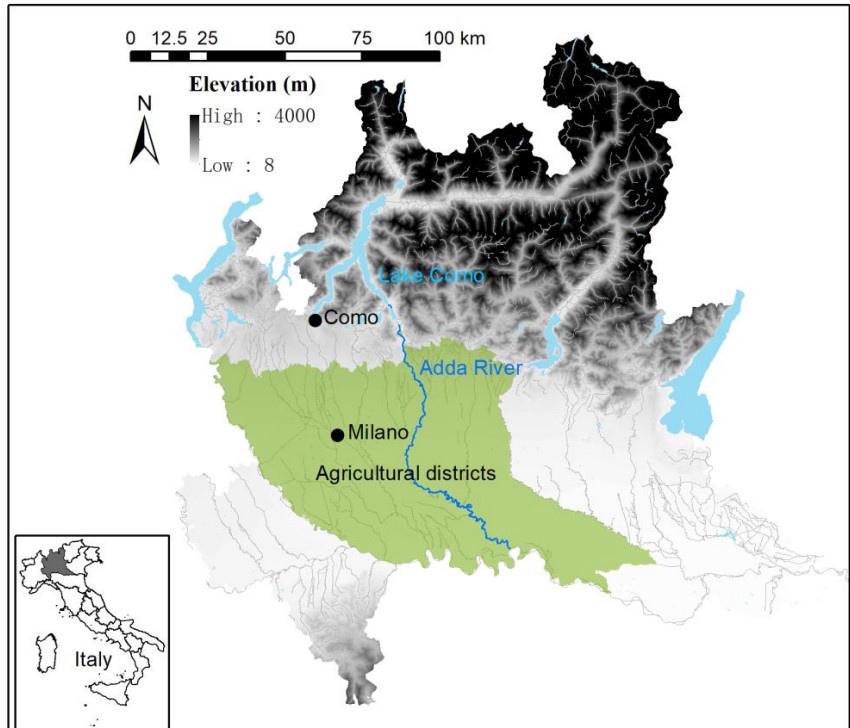

**Figure 1 Map of the Lake Como system in the Lombardy region, Northern Italy. The map was generated via Q-GIS using layers from the Geoportal of Regione Lombardia (http: //www.geoportale.regione.lombardia.it/, last access: July 2016).**




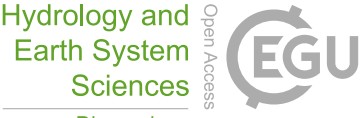



**Figure 2 Parallel coordinates plot of comparisons between: (a) all formulations, (b) traditional vs traditional & fair, (c) traditional vs inclusive, (d) traditional & fair vs inclusive, and (e) inclusive vs inclusive & fair. Each line connecting multiple axes represents one optimized solution, the horizontal dash line represents the mean line, and each type of line color represents one problem formulation. The axes with labels "Flood", "Irrigation", "Recreation", "Environment", and "Equity" refer to the corresponding performances of $J^F$, $J^I$, $J^R$, $J^E$, and $\zeta$, respectively, and the direction of solution preference is upward. Note: Figures (b), (c), (d), and (e) use the minimum and maximum performance values of Problems 1-4 as the lower and upper axis bounds, respectively.**







**Figure 3 Comparisons of the relationship between equity index and (a) mean $\mu$ and (b) standard derivation $\sigma$ of $J^F$, $J^I$, $J^R$, and $J^E$ for solutions from Problems P1-P4. Each dot in the scatter plots represents one optimal solution. Solutions with lower equity index $\zeta$ and higher performance mean value $\mu(J^F, J^I, J^R, J^E)$ are generally preferable. The arrow on y-axis shows the direction of preference.**




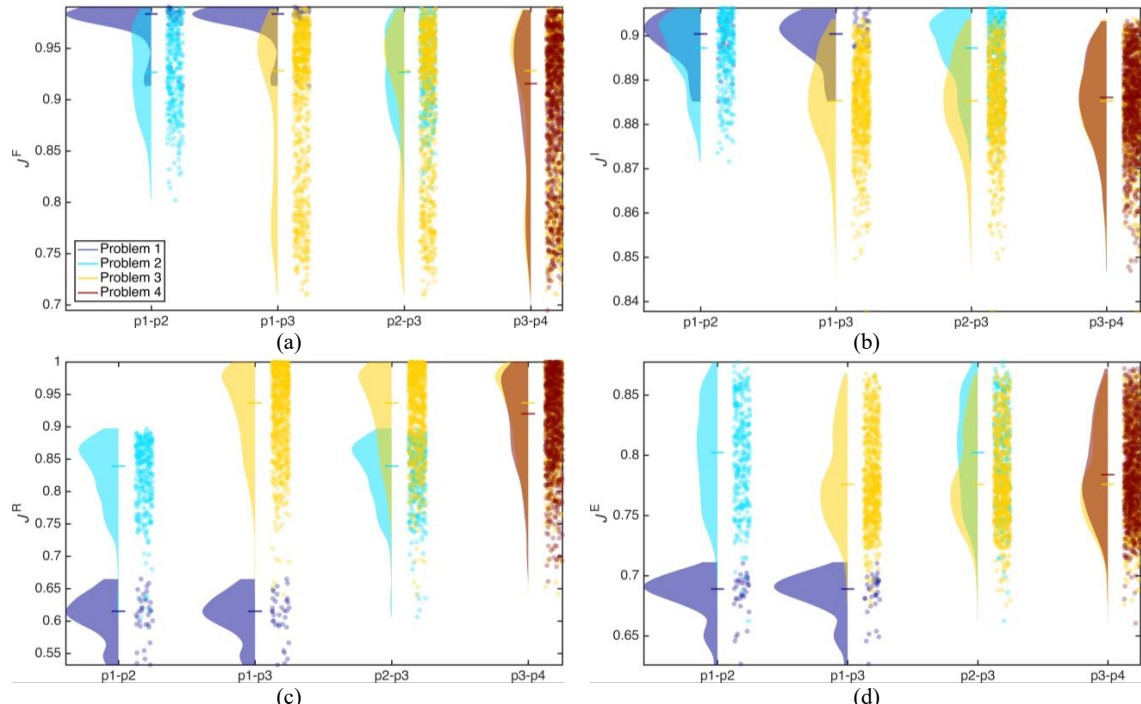

**Figure 4** Density distribution comparisons of (a) $J^F$, (b) $J^I$, (c) $J^R$, and (d) $J^E$ between problems P1 & P2, P1 & P3, P2 & P3, and P3 & P4. The density distribution is estimated by using kernel density estimation. Horizontal dash line refers to the median value of all solutions. The x-axis represents the comparison between various problem formulations.

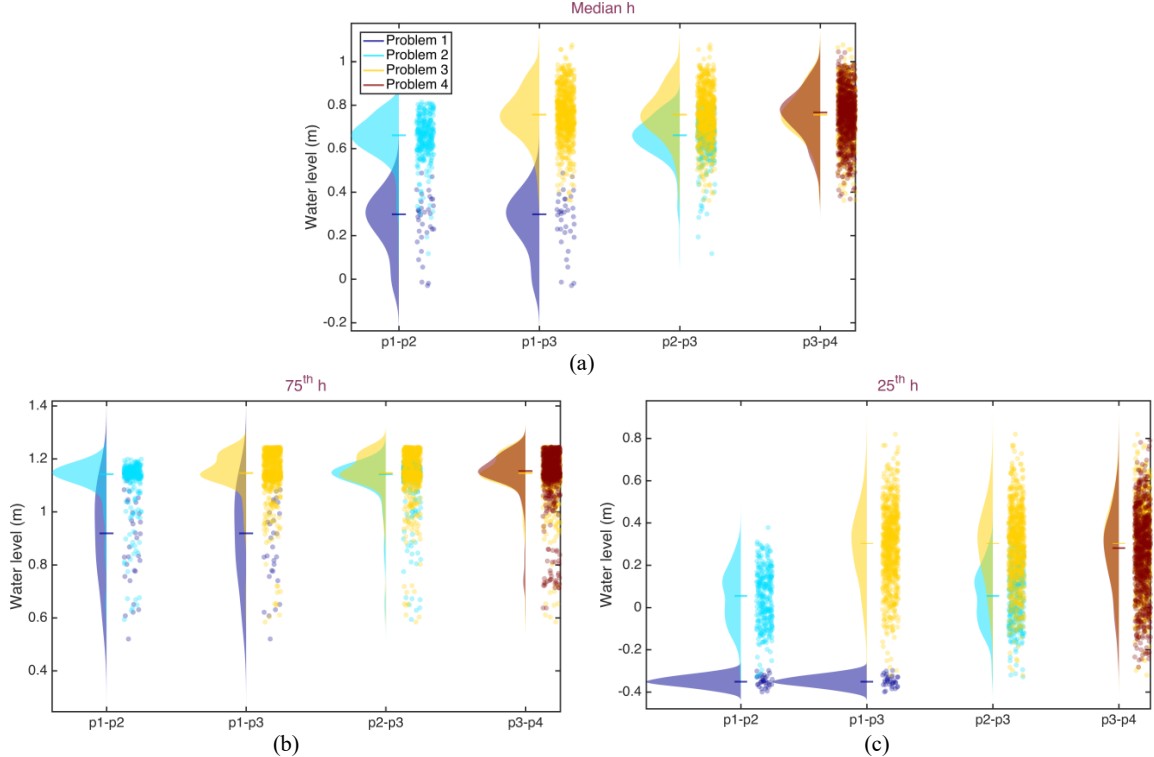

**Figure 5 Density distribution comparisons of (a) median, (b) 75th percentile, and (c) 25th percentile of lake water levels between problems P1 vs P2, P1 vs P3, P2 vs P3, and P3 vs P4. The density distribution is estimated by using kernel density estimation. The horizontal dash line refers to the median value of all solutions. The x-axis represents the comparison between various problem formulations.**

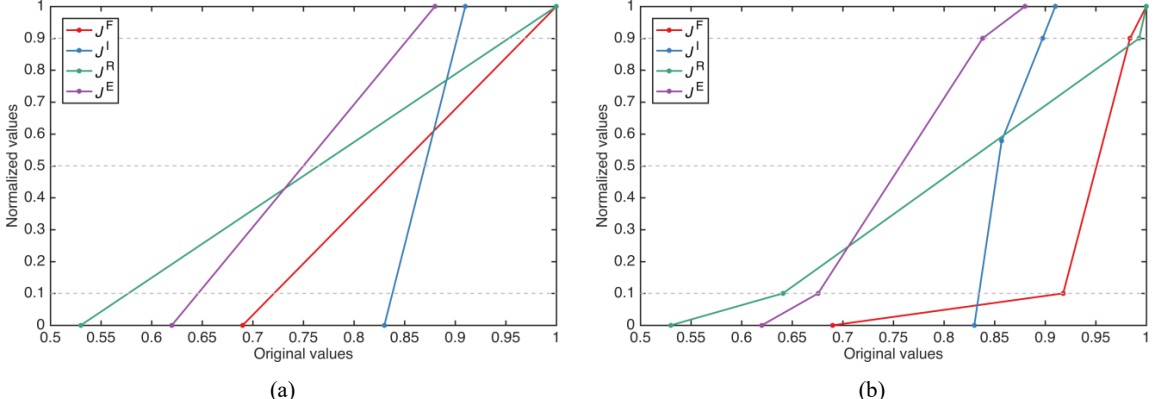

**Figure 6 Linear and non-linear value functions of $J^F$, $J^I$, $J^R$, and $J^E$ in the formulation of the equity index.**



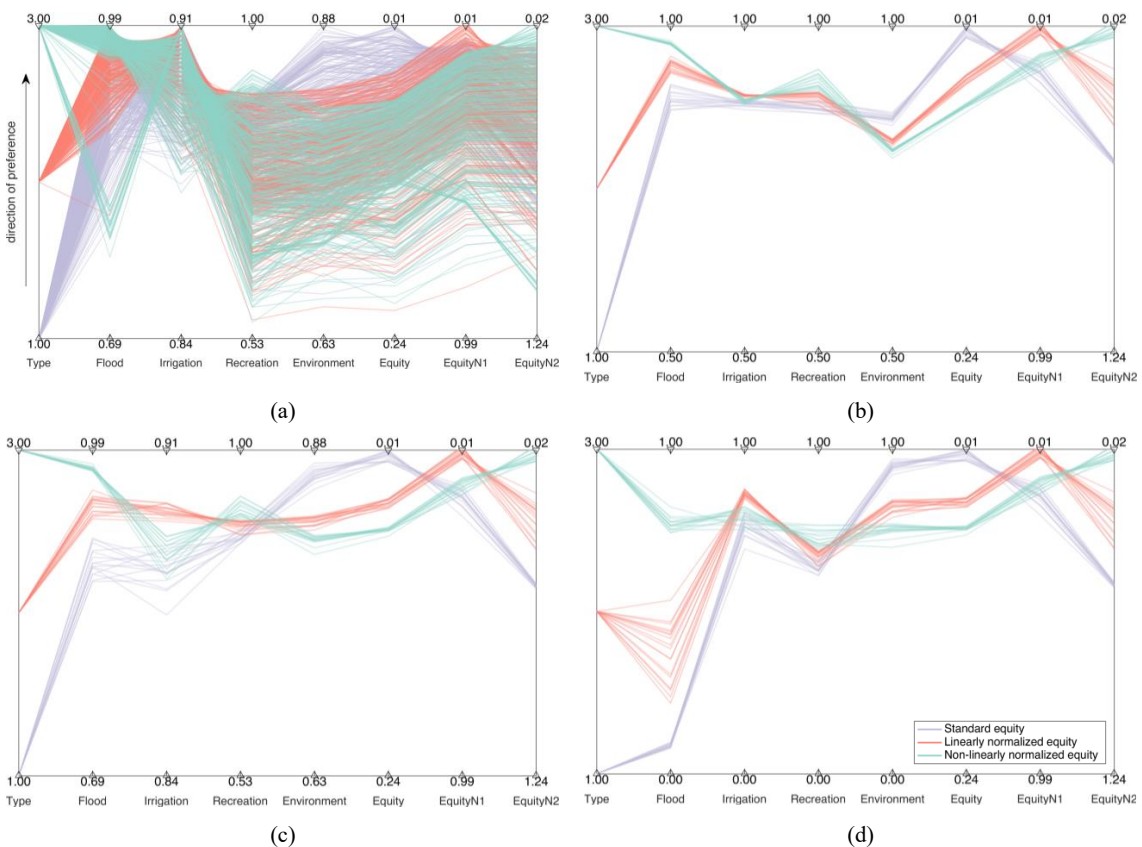

**Figure 7** Parallel plot of (a) all solutions and (b) (c) (d) 5% solutions with lowest (standard, linearly normalized, or non-linearly normalized) equity index in Problem 2. Type 1, 2, and 3 refer to optimizations minimizing standard, linearly normalized, and non-linearly normalized equity index, respectively. Figures (a), (b), and (c) show performance value, while figure (d) shows performance scores; Figures (b) and (d) have the same lower and upper bounds for all axes (of $J^F$, $J^I$, $J^R$, and $J^E$), while figures (a) and (c) use the minimum and maximum performance values of problems P1-P4 as the lower and upper axis bounds.