# Peer review of "Operationalizing equity in multipurpose water systems"

_Hydrology and Earth System Sciences, 2022_

## Author Comment (AC1)

[Figure]

**Figure A1.** *Number of compromise solutions for formulations F1-F4 that exceed increasing performance thresholds in all objectives.*

[Figure]

**Figure A2.** *Hypervolume indicator and best (transparent bar) and worst (solid bar) performance in each objective across the four formulations F1-F4.*

[Figure]

**Figure A3.** *Trajectories of average lake level for the best equitable solution in each problem formulation.*

[Figure]

**Figure A4**. Boxplot of best performance of 10 random optimization trials of F2, F3, and F3b, which refers to a new problem formulation including flood, irrigation, and environment.

*Revised equation (7).*

$$J^E = \frac{n_E}{H} \tag{1}$$

where $n_E$ is the number of days in the evaluation horizon $H$ during which $q_t^n - \sigma_t^n \leq r_{t+1} \leq q_t^n + \sigma_t^n$, with $q_t^n$ and $\sigma_t^n$ representing the mean and standard deviation, respectively, of the Adda river flow in natural conditions. It is worth noting that the ecosystem in the case study is sensitive to both high and low flows, thus a field or range instead of a minimum flow (which is typically used) to describe the ecosystem requirements.

---

## Author Response (AR1)

Reply to reviewers about the paper:

**Operationalizing equity in multipurpose water systems**

Guang Yang, Matteo Giuliani, and Andrea Castelletti

Matteo Giuliani, Assistant Professor
Department of Electronics, Information, and Bioengineering, Politecnico di Milano

Via Ponzio 34/5, 20133 Milano, Italy
Tel: +39 02 2399 9040
E-mail: matteo.giuliani@polimi.it

We thank the Editor and the 2 anonymous reviewers for their useful and constructive suggestions. We have carefully considered their concerns and revised the manuscript accordingly. In preparing the response to reviewers, we have used the following conventions: references to line numbers and figures are based on the revised manuscript (except if specifically mentioned); authors' replies are in blue; brief text additions are reported in red italics.

**REVIEWER #1**

1. This paper introduces equity metrics to design reservoir operations in multipurpose water resource problems. Equity metrics penalize variance across multiple metrics. The impact of introducing equity is evaluated through a set of four multi-objective formulations for the management of Lake Como, Italy. This is a pleasant read, with a clearly defined contribution that has the potential to be interesting. Yet, it is unclear from the paper what the introduction of an equity index can really bring to the resolution of a multipurpose reservoir operation problem. Authors need to address this in a revision to make this paper a significant contribution.

We thank the reviewer for the positive feedback and the detailed review. We have revised the paper accordingly and provided a point-by-point response to each comment below.

2. Authors note in the introduction that multi-objective optimization has gained traction in the field because it enables a posteriori decision-making (the decision is made with an understanding of the available alternatives, and the trade-offs involved) as opposed to a priori decision-making (where objectives are aggregated before alternatives are generated). An important remark in this context is that equity is by definition an a priori aggregation choice (as explored in 4.2). This means that this paper closes the circle by mixing a priori and a posteriori formulations. It is arguably the first to do it this way, but certainly not the first to explore this theme. Usually, these explorations are not groundbreaking, which is why water resource problem framings are either a priori or a posteriori, rather than a mix.

Then, what does adding equity really bring to the table? It seems this question is not asked beyond noting that it adds solutions, especially comparing P1 and P2. But the difference between P1 and P2 can hardly come as a surprise, as it is common for new solutions and trade-offs will emerge when a third objective is added to a two-objective problem.

We agree with the reviewer that the use of equity makes our approach a hybrid method that mixes features of a priori and a posteriori formulations, and this is a topic already explored in the literature. However, we argue our contribution goes beyond this point by showing how the operationalization of equity principles enriches the solution space by generating more compromise solutions than those obtained using a multi-objective optimization with traditional objective functions. This was probably not fully clear in the original manuscript - as noted also by Reviewer #2 - but the new figure below (which has been included in the revised manuscript) clearly shows the benefit of adding an equity objective to both the traditional (F1) and the inclusive (F3) formulations. Notably, there are 0, 47, 30, and 49 solutions having reliability greater than 0.85 in all objectives in F1, F2, F3, and F4, respectively, clearly demonstrating the benefit of including the equity index among the objective functions of the formulated problem. Interestingly, these compromise solutions account for 0%, 15.8%, 4.3%, and 6.1% of the total number of solutions obtained in each formulation, suggesting that formulation F2 is the most "efficient" in finding compromise solutions. This is also confirmed by observing how the slope of the

yellow (F3) and brown (F4) lines are steeper than the cyan one (F2), meaning that the number of compromise solutions in F3 decreases more evidently than in F2 for increasing performance thresholds.

[Figure]

***Figure A1.*** *Number of compromise solutions for formulations F1-F4 that exceed increasing performance thresholds in all objectives.*

We have included this figure (Figure 5) and the corresponding explanations in the revised manuscript as follows (see lines 242-250).

*To further understand the benefit of adding an equity objective to both the traditional (F1) and the inclusive (F3) formulations, in Figure 5 we compute the number of compromise solutions for each formulation considering only the policies that exceed increasing performance thresholds in all objectives. There are 0, 47, 30, and 49 solutions having reliability greater than 0.85 in all objectives in F1, F2, F3, and F4, respectively, clearly demonstrating how including the equity metric discovers favorable compromises that are not found by conventional formulations. Interestingly, these compromise solutions account for 0%, 15.8%, 4.3%, and 6.1% of the total number of solutions obtained in each formulation, suggesting that formulation F2 is the most "efficient" in finding compromise solutions. This is also confirmed by observing how the slope of the yellow (F3) and brown (F4) lines are steeper than the cyan one (F2), meaning that the number of compromise solutions in F3 decreases more evidently than in F2 for increasing performance thresholds.*

3. This is a shame because the paper has all the ingredients to present a compelling exploratory framework how to account for several objectives (here recreation / environment) through a single added objective. This is important because many-objective problems often stretch both the abilities of MOEAs and the available computational resources. Then, replacing several objectives with one could be extremely helpful. Here are the ingredients the paper has:

- The design P1 to P4 already enables to quantify the impact of (1) adding the objectives explicitly, (2) adding equity instead. Yet no metric is provided (hypervolume? Gain in selected objectives?) to provide this quantification.

Following the reviewer's comment, in the revised version of the paper, we have added a more quantitative assessment by including the new results illustrated in Figure A1 as well as the analysis of hypervolume and best-worst performance in each objective across the different formulations (see Figure A2 below). Results show that F3 and F4 have similar and the highest hypervolume. Apparently, non-dominated solutions obtained from F3 and F4 tend to be more efficient on $J^R$ and $J^E$, which are not explicitly optimized in F1 and F2. As also discussed in the paper, it is interesting to observe how formulation F2 attains a very good performance in the environment objective because the upper bound of $J^E$ is lower than other objectives, and thus the equity index tends to decrease (improve) as the value of $J^E$ increases.

[Figure]

***Figure A2.*** *Hypervolume indicator and best (transparent bar) and worst (solid bar) performance in each objective across the four formulations F1-F4.*

In the revised manuscript, we included this figure (Figure 4) and discussed the quantitative assessment as follows (see lines 236-241):

*To better quantify the differences in the solutions obtained with F1-F4, we computed the Hypervolume indicator along with the best and worst performance in each objective across the four formulations (see Figure 3). Results show that F3 and F4 have similar and the highest hypervolume, followed by F2, while F1 has much lower hypervolume. Apparently, non-dominated solutions obtained from F3 and F4 tend to be efficient on JR and JE, which are not explicitly optimized in F1 and F2. It needs to be noted that formulation F2 attains a very good performance in the environment objective because the upper bound of JE is lower than other objectives, and thus the equity index tends to improve as the value of JE increases.*

4.

- In Section 4.2, different ways to define equity are introduced. These should be presented in the experimental setup to explore (and quantify!) the impact of equity indicator choice.

Following the reviewer's comment, we have introduced the different formulations of the equity index in the revised Section 3 as follows (see lines 159-176):

*Yet, the ranges of performance attained by the different sets of solutions across the four objectives could be not identical. Using the equity index directly computed by aggregating the reliability performance across the four objectives may lead to some bias in the equity assessment, especially when the minimum and maximum performance values are significantly different. For example, if the ranges of $J^F$ and $J^E$ are [0.7, 1.0] and [0.6, 0.9], a fair solution having 0.85 reliability in all objectives would imply that $J^E$ would be highly satisfied as the 0.85 is close to the maximum value of 0.9; conversely, the performance in $J^F$ would be only intermediate as the 0.85 is still much lower than the maximum value of 1.0. To mitigate this bias, a value function can be used to first transform each objective into a satisfaction value expressed in a dimensionless scale (e.g., from 0 to 1), with the equity index computed by aggregating the satisfaction values as follows:*

$$\zeta' = \frac{\sigma[f^F(J^F), f^I(J^I), f^R(J^R), f^E(J^E)]}{\mu[f^F(J^F), f^I(J^I), f^R(J^R), f^E(J^E)]} \qquad (1)$$

*where $f^F(\cdot)$, $f^I(\cdot)$, $f^R(\cdot)$, and $f^E(\cdot)$ are the value functions of $J^F$, $J^I$, $J^R$, and $J^E$, respectively.*
*To explore the sensitivity of our results with respect to the use of different value functions, in this work, we examine the impacts of adopting both a linear and a non-linear value function (see Figure 2). The linear value function maps a performance value x to x' in the range [0, 1] where these extremes correspond to the lowest and highest values of reliability across all the solutions, respectively. In the non-linear value function, we hypothesized a function that requires high values of $J^F$ and $J^I$ to get high values of satisfaction in these objectives as the historical operation of the Lake Como is primarily looking at flood control and irrigation water supply while accepting lower levels of reliabilities in terms of $J^R$ and $J^E$. It should be noted that the purpose of this experiment is only testing the sensitivity of the equity index to the use of different values functions rather than capturing the preference structure of real stakeholders.*

5. Another important thing the revision could do is compare operations under different solutions (e.g., release and or lake levels as time series), to give a better sense of what solutions that are unique to one formulation bring to the set of alternatives that decision-makers could choose from.

Figure 5 in the original version of paper provides a summary of the system dynamics under different solutions by comparing the distributions of simulated lake levels. Following the reviewer's suggestion, in the revised version of the paper, we replaced that figure with Figure A3 below to better compare the trajectories of lake levels over time for the best equitable solution for F1-F4. Results show that the best equitable solution of F1 leads to the lowest water level especially during the late spring to reduce the flood risk and, at the same time, the water level drops significantly during the summer for better irrigation water supply, which are the only two objectives considered in this formulation. Conversely, the inclusive formulation F3 increases the lake level especially in the lake summer, as required to attain high performance in J^R. The water level under the equitable solutions of F2 and F4 is between that of F1 and F3, which indicates the identification of an alternative balancing the conflicting objectives.

[Figure]

***Figure A3.*** *Trajectories of average lake level for the best equitable solution in each problem formulation.*

In the revised manuscript we added the following text to discuss this new figure (see lines 264-270):

*Results in Figure 7 show that the best equitable solution of F1 leads to the lowest water level especially during the late spring to reduce the flood risk and, at the same time, the water level drops significantly during the summer for better irrigation water supply, which are the only two objectives considered in this formulation. Conversely, the inclusive formulation F3 increases the lake level especially in the late summer, as required to attain high performance in $J^R$. The water level under the equitable solutions of F2 and F4 is between that of F1 and F3, which indicates the identification of a compromise alternative balancing the conflicting objectives.*

6. Therefore, I would recommend for authors to demonstrate the interest of introducing equity indexes by enhancing their experimental design with (1) explicit consideration of different equity indexes at experimental stage, and (2) relevant metrics to quantify the differences between formulations. I would also encourage them to slightly revisit their introduction to note (i) that equity is an aggregation of several objectives, and (ii) that high numbers of objectives stretch our ability to solve multi-objective problems.

We thank the reviewer for these constructive suggestions. In addition to our replies above, we have further clarified in the revised manuscript introduction that equity is an aggregation of objectives that can also contribute in handling many objectives, which often stretches our ability to solve multi-objective optimization problems (see lines 74-76).

*Moreover, the aggregation of primary and marginalized objectives into an equity index makes our approach a hybrid method that blends a-posteriori decision making with an aggregated objective formulated a-priori, which can become particularly promising to stretch our ability to solve multi-objective problems with high numbers of objectives.*

**Miscellaneous minor remarks:**

7. Referencing all over the manuscript: please add single spaces between references when you have several at the same time (e.g., lines 85-86)

We have fixed this problem in the revised version of the paper.

8. Lines 42-44: Please add references on the efficiency / equity conflict in the water resource literature (a 1997 Loucks paper, and some of Ximing Cai's early paper, touch on that).

We have added the related references in the revised manuscript as below.

*This potential inconsistency between efficiency and equity might inadvertently bias the analysis on efficient but unfair solutions that the stakeholders will hardly accept (Cai et al., 2002; Cai et al., 2003; Cai, 2008; Loucks, 1997).*

*Cai, X. (2008). "Water stress, water transfer and social equity in Northern China—Implications for policy reforms." Journal of Environmental Management, 87(1), 14-25.*

*Cai, X., McKinney, D. C., and Lasdon, L. S. (2002). "A framework for sustainability analysis in water resources management and application to the Syr Darya Basin." Water Resources Research, 38(6), 21-1-21-14.*

*Cai, X., McKinney, D. C., and Rosegrant, M. W. (2003). "Sustainability analysis for irrigation water management in the Aral Sea region." Agricultural systems, 76(3), 1043-1066.*

*Loucks, D. P. (1997). "Quantifying trends in system sustainability." Hydrological Sciences Journal, 42(4), 513-530.*

9. Line 105: "could not be equal" denotes an impossibility. Is that what authors want to convey? If not, please rephrase.

No, we mean that the release r_t+1 might not be equal to the calculated decision u_t. We have rephrased it in the revised manuscript as below.

*The release $r_{t+1}$ does not necessarily equal the decision $u_t$ due to existing legal and physical constraints on the lake level and release (e.g., spills, dead storage).*

10. Equation (1): could authors please say a word on neglecting other terms (in particular evaporation from the lake)?

Actually in equation 1 we used net inflow that allows accounting for the evaporation from the lake. In the revised version of the paper, we have clarified this point as below.

*where $q_{t+1}$ ($m^3$/s) and $r_{t+1}$ ($m^3$/s) are the net inflow (i.e., inflow minus evaporation losses) to the lake and the actual lake release in the time period [t, t+1), respectively.*

11. Line 111: ":" missing at the end of the sentence; the reference should not come with extra "()"

We have fixed this in the revised manuscript as below.

*The lake operating policies that determine the release decision $u_t$ are defined as Gaussian radial basis functions (RBFs; Buşoniu et al. (2011)) as follows:*

12. Line 116: why is 4 an appropriate number of RBFs in this case?

The number of RBFs are generally determined by sensitivity analysis, i.e., increasing the number of RBFs until the optimization performance (e.g., hypervolume of the solutions) does not change significantly. A systematic analysis suggesting the use of 4 RBFs for a single reservoir has been conducted by Giuliani et al. 2016, and this number has been confirmed to work well also in the case of Lake Como where it was used in previous works (e.g. Denaro et al., 2017; Giuliani et al. 2020). We have clarified this point in the revised version of the paper as below.

*the number of RBFs is set to four (K=4) which proves effective in our previous works (Giuliani et al., 2016b; Giuliani et al., 2020).*

Giuliani, M., A. Castelletti, F. Pianosi, E. Mason, and P. Reed (2016), Curses, tradeoffs, and scalable management: advancing evolutionary multi-objective direct policy search to improve water reservoir operations, Journal of Water Resources Planning and Management, 142(2)

Denaro, S., D. Anghileri, M. Giuliani, and A. Castelletti (2017), Informing the operations of water reservoirs over multiple temporal scales by direct use of hydro-meteorological data, Advances in Water Resources, 103, 51–63

Giuliani, M., L. Crochemore, I. Pechlivanidis, and A. Castelletti (2020), From skill to value: isolating the influence of end user behavior on seasonal forecast assessment, Hydrology and Earth System Sciences, 24

13. Section 3.2: my advice would be to use subscripts rather than superscripts, in particular because equation (4) parameter n^F reads like an exponent (same for equation (6)).

We thank the reviewer for the suggestions. We have changed the superscripts to subscripts in the revised version of the paper.

14. Equation (7): please define all reliabilities in a consistent format. Besides, the introduction of an upper environmental flow limit, while great to see, requires a justification in a field where ecosystem requirements are classically represented with a minimum flow only.

We have redefined the reliability of environmental flow in equation 7 to keep the consistent format as below.

$$J^E = \frac{n_E}{H} \qquad\qquad (2)$$

*where $n_E$ is the number of days in the evaluation horizon H during which $q_t^n - \sigma_t^n \leq r_{t+1} \leq q_t^n + \sigma_t^n$, with $q_t^n$ and $\sigma_t^n$ representing the mean and standard deviation, respectively, of the Adda river flow in natural conditions.*

It is worth noting that the ecosystem in the case study is sensitive to both high and low flows and the thresholds vary with season. That is the reason why we use a range instead of a minimum flow to describe the ecosystem requirements. We have further explained this in the revised manuscript as below (see lines 144-147):

*It is worth noting that the ecosystem in the case study is sensitive to both high and low flows, thus the target of maintaining the lake release within a range approximating the natural variability instead of considering only a minimum flow threshold as traditionally done in the literature.*

15. Please choose between "formulation" and "problem" to describe the alternative framings throughout (with a personal preference for "formulation" with codes F1 to F4, but authors to decide).

We thank the reviewer for the suggestions and have used 'formulation' with codes F1-F4 throughout the manuscript.

16. Figure 2: please include the color code in the legend, and / or add P1 to P4 as labels on the left y-axis (instead of using floats: there is no problem 1.00 or 4.00, and this is all very confusing, as is the absence of correspondence on the figure between formulations and PX numbers). Finally, it is not clear the vertical scaling should be different in panel (a) compared with all other panels.

We have added F1 to F4 as labels on the left y-axis. We use the different vertical scaling in panel (a) with other panels in order to highlight the different upper and lower bounds of each objective. In addition, the y-axis of each objective has the same maximum (1.00) and minimum (0.50) limits in panel (a), which makes compromised solutions (showing with lines approximately horizontal) more conspicuous. We have explained this in the revised caption of Figure 2 as below.

*Note: Panel (a) uses the same lower and upper axis bounds for each objective to better compare their best and worst performances and discover compromised solutions (showing with lines approximately horizontal), while Panels (b), (c), (d), and (e) use the minimum and maximum performance values of Problems 1-4 as the lower and upper axis bounds, respectively.*

17. Section 4.2 Why not make that part of the experimental design in Section 3? This would make this section much simpler to follow.

We thank the reviewer for the suggestion and have moved the experiment design of Section 4.2 to Section 3 (see our response to point 4).

**REVIEWER #2**

1. This paper explores the impact of including an equity measure during multiobjective search for reservoir operating policies in the Lake Como system. Using four rival framings of the search problem, the authors show how including equity as an objective impacts the Pareto approximate set. The authors further explore the sensitivity of solutions discovered through each formulation to the definition of the equity measure. This paper is well written and within the scope of this journal. I believe it will represent a significant contribution to the field after the authors undertake revisions to improve the clarity of their results and contextualize the impact of their findings. My main comments are listed below:

Thank you very much for your positive feedback. We have revised the paper and provided a point-by-point tentative response to each comment.

2. My main concern with the current submission is that the results don't clearly highlight to potential benefits of including the equity metric for the higher dimensional formulation of the problem. The authors assert that adding the equity indicator generates more solutions that mitigate conflicts between conflicting objectives (lines 245-249). While this is clearly the case when comparing problem framings 1 and 2 (the "traditional" and "traditional and fair" framings), the difference between framings 3 and 4 ("inclusive" and "inclusive and fair") is hard to assess from Figure 2e. The number of solutions plotted in Figure 2 makes it hard to distinguish the differences between formulations 3 and 4, and it is all but impossible to see trade-offs between conflicting objectives. The paper would benefit from additional visualizations that highlight compromise solutions discovered using each formulation and demonstrate how (or if) including the equity metric discovers favorable compromises that are not found by other formulations. One way to do this could be to filter the Pareto approximate set according to a set of desired performance criteria for each objective and compare how many solutions each formulation discovers that meet the criteria.

We thank the Reviewer for highlighting this point, and we agree that additional visualizations will be beneficial as the difference between framings 3 and 4 ("inclusive" and "inclusive and fair") is hard to assess from Figure 2e. To further illustrate the advantage of the fair formulation (problem framings 2 and 4) in obtaining (relatively more) compromise solutions, we followed the reviewer's comment and filtered the Pareto approximate set considering only solutions that exceed a desired performance threshold in all objectives.

[Figure]

**Figure A1.** *Number of compromise solutions for formulations F1-F4 that exceed increasing performance thresholds in all objectives*

Notably, there are 0, 47, 30, and 49 solutions having reliability greater than 0.85 in all objectives in F1, F2, F3, and F4, respectively. These compromise solutions account for 0%, 15.8%, 4.3%, and 6.1% of the total number of solutions obtained in each formulation, suggesting that formulation F2 is the most "efficient" in finding compromise solutions. This is also confirmed by observing how the slope of the yellow (F3) and brown (F4) lines are steeper than the cyan one (F2), meaning that the number of compromise solutions in F3 decreases more evidently than in F2 for increasing performance thresholds.

We believe that this new figure can clarify how including the equity metric allow the discovery of favorable compromises that are not found by the traditional and inclusive formulations.

We have included this figure (Figure 5) and the corresponding explanations in the revised manuscript as follows (see lines 242-250).

*To further understand the benefit of adding an equity objective to both the traditional (F1) and the inclusive (F3) formulations, in Figure 5 we compute the number of compromise solutions for each formulation considering only the policies that exceed increasing performance thresholds in all objectives. There are 0, 47, 30, and 49 solutions having reliability greater than 0.85 in all objectives in F1, F2, F3, and F4, respectively, clearly demonstrating how including the equity metric discovers favorable compromises that are not found by conventional formulations. Interestingly, these compromise solutions account for 0%, 15.8%, 4.3%, and 6.1% of the total number of solutions obtained in each formulation, suggesting that formulation F2 is the most "efficient" in finding compromise solutions. This is also confirmed by observing how the slope of the yellow (F3) and brown (F4) lines are steeper than the cyan one (F2), meaning that the number of compromise solutions in F3 decreases more evidently than in F2 for increasing performance thresholds.*

3. The paper could be strengthened by including a deeper discussion of how equity fits into the decision context of the management problem. The methodology proposed in this paper serves to facilitate the discovery of equitable compromise policies - but will these policies be examined by a social planner balancing the trade-offs across multiple performance interests, or serve as the basis for a negotiation process where multiple stakeholders must find an acceptable compromise? Could this methodology be extended to systems with multiple stakeholders with similar performance objectives, and how would that change the application?

We thank the reviewer for this suggestion which indeed deserves to be better discussed. As the reviewer says, our methodology aims at facilitating the design of equitable compromise policies, and it is demonstrated using the Lake Como as a case study where the lake is operated by a single authority (i.e. Consorzio dell'Adda), which somehow acts as a social planner that first analyzes the trade-offs across the interests of multiple stakeholders and then implements a selected compromise policy. However, the same approach can also serve as the basis for an interactive negotiation with multiple stakeholders that can discuss and analyze the same set of solutions examined by the social planner in order to find an acceptable compromise. It is important to stress that in both cases, however, the formulation of the equity index should be co-designed with the stakeholders through the identification of suitable value functions to map the original objectives into satisfaction values that allow the aggregation of the originally incommensurable objectives into an equity index. We have discussed this point better in the revised conclusions section of the paper as below (see lines 318-325):

*Our methodology is demonstrated using the Lake Como as a case study where the lake is operated by a single authority (i.e. Consorzio dell'Adda), which somehow acts as a social planner that first analyzes the trade-offs across the interests of multiple stakeholders and then implements a selected compromise policy. However, the same approach can also serve as the basis for an interactive negotiation with multiple stakeholders that can discuss and analyze the same set of solutions examined by the social planner in order to find an acceptable compromise. It is important to stress that in both cases, however, the formulation of the equity index should be co-designed with the stakeholders through the identification of suitable value functions to map the original objectives into satisfaction values that allow the aggregation of the originally incommensurable objectives into an equity index.*

4. I find it confusing that P2 outperforms P3 in the environmental objective. While lines 225-228 explain that the lower performance of the environmental objective from P1 incentivizes high values of the environmental objective, I have a hard time believing that this sends a stronger signal than directly optimizing for the objective. Could this indicate a search failure in P3?

This is an interesting point raised by the reviewer because it is true that directly optimizing for one objective should get the solution that performs best in this objective. However, in multi-objective optimization, the size of the non-dominated space grows exponentially with the number of objectives (Farina and Amato 2002). Convergence difficulties have been found in many multi-objective evolutionary algorithms (MOEAs) when solving problems having four or more objectives (Corne and Knowles 2007; Hughes 2005; Purshouse and Fleming 2007) because when the number of objectives increases, a large population size will be needed to represent the resulting Pareto-optimal front and the evaluation of diversity measure will become computationally expensive. Although some algorithms, such as Borg MOEA (Hadka and Reed 2013) used in this study, have been shown to be more efficient and reliable in solving a variety of multi and many-objective problems, there are no guarantees that they can always discover the real optimal solutions.

In our experiments, each problem formulation was solved by running 10 independent optimization trials with 2 million function evaluations. This set-up was identified as reliable for the Lake Como case study in previous works. The difference in the best environment performance between F2 and F3 is likely attributable to the increasing challenges introduced by the additional objectives considered in this work. Running a more extensive optimization will likely cover this gap by requiring substantial computational efforts, but this will not change the main contributions of the paper. However, looking at the results of the 10 optimization trials (see Figure A4 below, which has been added in the Appendix as a Supplementary Figure), we can notice how the median of the best performance of F2 and F3 is indeed almost equivalent. A Kolmogorov-Smirnov test confirmed the best $J^E$ performance from the two formulations are from the same statistical distribution at 1% significance level. Moreover, the results of a new problem formulation F3b including flood, irrigation, and environment (i.e., removing the recreation objective from F3 to have the same number of objectives as F2) show that F3b can obtain better performance than F2 by directly optimizing the environment. However, including in F3 an additional and strongly conflicting objective (recreation) makes the search for the best environment more challenging.

[Figure]

Figure A4. Boxplot of best performance of 10 random optimization trials of F2, F3, and F3b, which refers to a new problem formulation including flood, irrigation, and environment.

We have discussed this aspect in the revised version of the manuscript as follows (see lines 228-235):

*It needs to be noted that F3 which directly includes $J^E$ should theoretically outperform F2 in the environmental objective. The difference in the best environment performance between F2 and F3 is likely attributable to the increasing challenges introduced by the additional objectives considered in this work. According to the boxplot of best performance of 10 random optimization trials reported in Supplementary Figure S2, the median of the best performance of F2 and F3 is indeed almost equivalent. Moreover, a new problem formulation F3b including flood, irrigation, and environment (i.e., removing the recreation objective from F3 to have the same number of objectives as F2) can obtain better performance than F2 by directly optimizing the environment. Including in F3 an additional and strongly conflicting objective (recreation) makes the search for the best environment more challenging.*

**Minor comments:**

1. I would recomend including the following papers in the review of equity in water resources research (lines 44-59):

Osman, K. K., & Faust, K. M. (2021). Toward operationalizing equity in water infrastructure services: Developing a definition of water equity. ACS ES&T Water, 1(8), 1849-1858.

Jafino, B. A., Kwakkel, J. H., Klijn, F., Dung, N. V., van Delden, H., Haasnoot, M., & Sutanudjaja, E. H. (2021). Accounting for multisectoral dynamics in supporting equitable adaptation planning: A case study on the rice agriculture in the Vietnam Mekong Delta.Earth's Future 9(5), e2020EF001939.

Fletcher, S., Hadjimichael, A., Quinn, J., Osman, K., Giuliani, M., Gold, D., ... & Gordon, B. (2022). Equity in water resources planning: a path forward for decision support modelers. Journal of Water Resources Planning and Management. 148(7), 02522005.

Thank you for these suggestions, which we have included in the revised version of the paper as follows.

*Equity here is defined as "the provision of a consistent minimum quality and quantity, determined at the local level, of water services to all end-users" (Osman and Faust, 2021).*

*..., while including equity among the objectives can be useful to ensure that the negotiations on the solution to be implemented succeed smoothly (Jafino et al., 2021).*

*There is a growing interest in equity-related research in the water resources literature, with Fletcher et al. (2022) recently offering some actionable recommendations about the integration of equity into the water resources planning*

2. I don't find the trade-offs between J_R and J_F/J_I under P3 to be "remarkable" (line 194). To me, these trade-offs make sense and would be expected. Non-dominated solutions in P3 include solutions that maximize J_R, which may come at the expense of J_I and J_F. P2 does not have this incentive. Though the equity objective incentivizes J_R more than P1, it does not do so at the cost of other objectives.

Thank you for the remark. We agree that these tradeoffs are expected, and we actually meant that more solutions with an extremely high level of J_R (and low level of J_I and J_F) are obtained under F3 than F2 (figure 2d). In the revised version of the manuscript, we have rephrased the sentence accordingly as below.

*The inclusive optimization supports the full exploration of the trade-offs between the four objectives, remarkably improving $J^R$ but degrading the performance in terms of $J^F$ and $J^I$ (i.e., the maximum performance in $J^R$ is equal to 1, while the worst solution in flood and irrigation supply is much lower than with F1 or F2).*